# Associations of Tinnitus Incidence with Use of Tumor Necrosis Factor-Alpha Inhibitors among Patients with Autoimmune Conditions

**DOI:** 10.3390/jcm12051935

**Published:** 2023-03-01

**Authors:** Nirvikalpa Natarajan, Shelley Batts, Saurabh Gombar, Raj Manickam, Varun Sagi, Sharon G. Curhan, Konstantina M. Stankovic

**Affiliations:** 1Department of Otolaryngology—Head and Neck Surgery, Stanford University School of Medicine, 801 Welch Rd, Palo Alto, CA 94304, USA; 2Atropos Health, 71 W 83rd St #3R., New York, NY 10024, USA; 3Department of Pathology, Stanford University School of Medicine, 300 Pasteur Drive, Stanford, CA 94305, USA; 4Channing Division of Network Medicine, Brigham and Women’s Hospital and Harvard Medical School, 181 Longwood Avenue, Boston, MA 02115, USA; 5Department of Neurosurgery, Stanford University School of Medicine, 801 Welch Rd, Palo Alto, CA 94304, USA; 6Wu Tsai Neuroscience Institute, Stanford University, 288 Campus Dr, Stanford, CA 94305, USA

**Keywords:** autoimmune disorders, cohort study, electronic health records, incidence, propensity score matching, tinnitus, tumor necrosis factor-alpha, tumor necrosis factor-alpha inhibitor

## Abstract

Tumor necrosis factor-alpha (TNFα) may promote neuroinflammation prompting tinnitus. This retrospective cohort study evaluated whether anti-TNFα therapy influences incident tinnitus risk among adults with autoimmune disorders and no baseline tinnitus selected from a US electronic health records database (Eversana; 1 January 2010–27 January 2022). Patients with anti-TNFα had ≥90-day history pre-index (first autoimmune disorder diagnosis) and ≥180-day follow-up post-index. Random samples (*n* = 25,000) of autoimmune patients without anti-TNFα were selected for comparisons. Tinnitus incidence was compared among patients with or without anti-TNFα therapy, overall and among at-risk age groups or by anti-TNFα category. High-dimensionality propensity score (hdPS) matching was used to adjust for baseline confounders. Compared with patients with no anti-TNFα, anti-TNFα was not associated with tinnitus risk overall (hdPS-matched HR [95% CI]: 1.06 [0.85, 1.33]), or between groups stratified by age (30–50 years: 1 [0.68, 1.48]; 51–70 years: 1.18 [0.89, 1.56]) or anti-TNFα category (monoclonal antibody vs. fusion protein: 0.91 [0.59, 1.41]). Anti-TNFα was not associated with tinnitus risk among those treated for ≥6 months (hdPS-matched HR [95% CI]: 0.96 [0.69, 1.32]) or ≥12 (1.03 [0.71, 1.5]), or those with RA (1.16 [0.88, 1.53]). Thus, in this US cohort study, anti-TNFα therapy was not associated with tinnitus incidence among patients with autoimmune disorders.

## 1. Introduction

Tinnitus, the perception of sound in the absence of an external source, affects an estimated 14% of people worldwide, with ~2% experiencing severe symptoms [1]. Tinnitus presents with hearing loss in the majority (>90%) of cases [2,3] and the risk increases with age [4,5]. The clinical impact can be substantial and potentially disabling, as chronic tinnitus is associated with higher levels of anxiety, depression, irritability, sleep disturbances, and stress, as well as negative impacts to quality of life [3,6]. However, there is currently no cure for tinnitus and the limited existing treatments are associated with low efficacy and heterogenous response [7]. Given the substantial clinical and humanistic burden of tinnitus, there is an urgent unmet need for effective medical therapies. However, progress towards the development of such therapies is stymied by the lack of a detailed understanding of tinnitus’ etiology, although the general consensus is that cochlear damage triggers aberrant activity and encoding in higher auditory processing centers (e.g., the auditory cortex [AC] and inferior colliculus [IC]) [8,9].

Neuroinflammation may play a role in the development of tinnitus, possibly due to cochlear damage following the release of inflammatory cytokines and chemokines in response to acoustic trauma or systemic inflammation [10,11,12,13]. In animal models, noise exposure prompted an inflammatory response along the central auditory pathway, resulting in behavior consistent with tinnitus and increased expression of proinflammatory cytokines, particularly tumor necrosis factor-alpha (TNFα) [14]. TNFα expression is also significantly upregulated in the IC or AC of mice with salicylate-induced tinnitus [15,16]. Further, in mice, intra-ventricle infusion of recombinant TNFα resulted in tinnitus behavior and microglia alterations in the AC following noise exposure [17], and genetic knockout of TNFα or blockade of TNFα expression decreased tinnitus-associated behavior [14,18,19]. Finally, intracochlear TNFα infusion resulted in reduced auditory nerve activity and degradation of inner hair cell synapses in guinea pigs, which could be prevented via systemic TNFα blockade [20].

TNFα has been implicated in some exploratory human studies of tinnitus. Certain polymorphisms in the *TNF* gene are associated with susceptibility to tinnitus among older individuals with occupational noise exposure [21]. In patients with chronic tinnitus, relaxation training that significantly decreased tinnitus-related stress, depression, anger, and disturbance was associated with reduced serum TNFα levels [22]. Serum levels of TNFα have been positively associated with tinnitus loudness, stress, and depression in some studies [23], but not others [24].

TNFα is involved in the pathogenesis of multiple inflammatory or autoimmune diseases [25]. As of 2022, five TNFα inhibitors (anti-TNFα) have been approved by the US Food and Drug Administration and are widely prescribed for rheumatoid arthritis (RA), psoriatic arthritis, plaque psoriasis, Crohn’s disease (CD), ankylosing spondylitis (AS), ulcerative colitis (UC), and noninfectious uveitis (NIU) [26]. These include etanercept, a dimeric human recombinant fusion protein (FP), and the chimeric or fully humanized monoclonal antibodies (AB) infliximab, adalimumab, certolizumab pegol, and golimumab.

Although some evidence suggests a link between TNFα and tinnitus genesis, it is unknown whether anti-TNFα therapy influences the development of tinnitus. Therefore, the aim of the present study is to evaluate whether the incidence of tinnitus among individuals with autoimmune conditions for which anti-TNFα are indicated differs between those who were and were not treated with anti-TNFα therapy, using a US electronic health records (EHR) database.

## 2. Methods

### 2.1. Study Design and Data Source

This US population-based retrospective cohort study compared the incidence of tinnitus among adults (aged 18 to 89 years) diagnosed with RA, psoriasis, UC, CD, AS, or NIU who were or were not treated with an FDA-approved anti-TNFα therapy (infliximab, adalimumab, certolizumab pegol, golimumab, or etanercept). We used a nationally representative EHR dataset (1 January 2010 to 27 January 2022) from Eversana Life Sciences. The EHR data represent all regions of the continental 48 US states and comprises community hospitals and large provider practices, capturing inpatient, outpatient, emergency room, and urgent care encounters. During the study period, the total sample size of the EHR database was approximately 38 million patients.

Data elements extracted for analysis included demographics (age, sex, race/ethnicity); diagnostic, procedure, and treatment codes; numbers of database encounters; and comorbidity profiles (Charlson Comorbidity Index [CCI] score and individual CCI disorders [27]). Autoimmune disorder and tinnitus diagnoses were identified based on International Classification of Diseases, 9th/10th editions codes, and anti-TNFα therapies were identified with Anatomical Therapeutic Chemical codes (Appendix A).

### 2.2. Study Populations and Study Periods

#### 2.2.1. Overall Tinnitus Prevalence among All Adults

The overall prevalence of tinnitus (2010–2021) was estimated among adults aged 18 to 89 years in the EHR database with a diagnostic code for tinnitus (Appendix A).

#### 2.2.2. Incidence of Tinnitus among Adults with Autoimmune Disorders, Who Did or Did Not Receive Anti-TNFα 

The index date was defined as the first diagnosis of an autoimmune disorder. Adults aged 18 to 89 years with (1) a diagnosis of RA, psoriasis, AS, UC, CD, or NIU at baseline; (2) no diagnosis of tinnitus during the baseline period; and (3) ≥90-day history pre-index and ≥180-day history post-index in the database (autoimmune cohort) were included in the analyses of incident tinnitus. Patients were followed from index to diagnosis of tinnitus, death, loss to follow-up, or data end, whichever came first. A patient could have more than one diagnosis and there could be multiple diagnoses in each encounter with the health system.

For the analysis of incident tinnitus according to use of anti-TNFα therapy, the autoimmune cohort was further defined into subcohorts who did (Yes-TNFα cohort) or did not (No-TNFα cohort) receive anti-TNFα therapies at baseline or during the study period. Given the large sample size of the No-TNFα cohort, 25,000 randomly selected patients were sampled for analysis and propensity score (PS) matching such that it was sufficiently large to capture confounders but manageable for analysis. Additional cohorts were structured by two age groups (30–50 and 51–70 years), selected due to the higher likelihood of presbycusis and related tinnitus [5,28].

For the analyses of incident tinnitus by anti-TNFα therapy type, patients meeting the criteria for the Yes-TNFα cohort were required to have an indication regarding the type of anti-TNFα received. Cohorts were defined by use of anti-TNFα AB (infliximab, adalimumab, certolizumab pegol, and golimumab; TNFα-AB cohort) or FP (etanercept; TNFα-FP cohort) during the baseline or study periods. Additional cohorts were constructed of patients with anti-TNFα use for ≥6 and ≥12 months (i.e., ≥2 codes for anti-TNFα spanning those timeframes).

### 2.3. Outcomes

The main outcome was the rate of incident tinnitus during the study period among patients with autoimmune disorders who did or did not receive anti-TNFα (detailed below). A secondary outcome was the prevalence of tinnitus among the entire adult population in the EHR database. Demographic information (i.e., age, sex, race/ethnicity, number of encounters, comorbidities, and year of cohort entry) was collected and time to tinnitus diagnosis from index was assessed.

#### Handling of Missing Data

The handling of missing data depended on the type of outcome variable. For binary variables, a missing value was replaced with zero. Missing values in continuous variables were dropped. The Ns selected for the analyses were 25,000 for all no-TNFα groups but when analyzing the outcome of tinnitus, the program may have encountered missing values that were excluded from analysis.

### 2.4. Covariates and PS Matching

In the comparisons of tinnitus incidence, cohorts were matched using PSs, which provide a composite score of the baseline confounders such that when the PS is balanced (within a caliper of 0.25) between arms, their baseline confounders would also become balanced [29]. Potential confounders (covariates) included all collected baseline demographic information with standardized mean difference (SMD) > 0.25 between the two comparative cohorts. Baseline confounders were computed using diagnostic codes observed in the 90 days pre-index and further classified using CCI disorders (Appendix A).

Patients from one cohort were matched 1:1 with patients in the comparator cohort using two algorithms: (1) basic matching on age and sex; and (2) high-dimensionality PS (hdPS) matching [30] using age, sex, CCI score [31], diagnostic codes, procedure codes, medication codes, and number of encounters in the EHR database. To compute the hdPS score, hdPS covariates were first generated from diagnostic and treatment codes as described in Schneeweiss et al. [30]. A PS model of the hdPS covariates was then fitted using logistic regression with Least Absolute Shrinkage and Selection Operator (LASSO) regularization that penalizes low weight (i.e., less contributory) variables down to zero weights, such that the resulting parsimonious model has equivalent predictive performance without overfitting too many covariates in a high-dimensional setting [32,33]. The LASSO hyperparameter was tuned using 5-fold cross-validation and the 1-standard error rule [34].

### 2.5. Statistical Analyses

Descriptive statistics were reported as means and standard deviations (SD) for continuous variables and as frequencies and proportions for categorical variables. The incidence of tinnitus was compared between cohorts who were or were not treated with anti-TNFα therapies, overall, by the type of anti-TNFα therapy, and those aged 30–50 and 51–70 years. Sensitivity analyses were conducted to assess the incidence of tinnitus among patients with (1) a diagnosis of RA, and (2) ≥6 months or (3) ≥12 months of anti-TNFα therapy. Hazard ratios (HRs) with 95% confidence intervals (CIs) were computed and reported in the unadjusted, basic-matched (age/sex), and hdPS-matched datasets. Time-to-event analysis using Cox proportional hazards regression was used to determine the time (days) from the first autoimmune disease diagnosis to first tinnitus diagnosis or data end.

A two-sided *p* value of 0.05 denoted statistical significance. All analyses were performed using R (v4.2.1, R Core Team 2022) using the Atropos Health real-world evidence platform.

## 3. Results

### 3.1. Sample Selection

#### 3.1.1. Tinnitus Prevalence Cohorts

Of 28,387,160 patients in the EHR database (2010–2021), 155,091 had a diagnosis of tinnitus at any timepoint, yielding an overall prevalence of 0.55%.

#### 3.1.2. Tinnitus Incidence Cohorts

For the analysis of incident tinnitus among patients with autoimmune disorders at baseline, 13,293 patients with anti-TNFα therapy (Yes-TNFα) and a random sample of 25,000 patients with no anti-TNFα therapy (No-TNFα) were selected (Appendix A). The Yes-TNFα cohort was further stratified by age 30–50 years (*n* = 4397) and 51–70 years (*n* = 6868) for separate comparisons with 25,000 randomly selected, similarly aged patients without anti-TNFα therapy.

For the analyses of incident tinnitus by category of anti-TNFα, 2397 and 9471 patients with autoimmune disorders were selected to the TNFα-FP and TNFα-AB samples, respectively. Separate comparisons of tinnitus incidence were conducted between the TNFα-FP (*n* = 3506) and TNFα-AB (*n* = 10,859) cohorts and a random sample of 25,000 patients who did not receive anti-TNFα therapy.

For the sensitivity analyses, 4733 and 3516 patients had ≥6 and ≥12 months duration of anti-TNFα therapy, respectively, and 6824 had RA and used anti-TNFα therapy.

### 3.2. Demographic and Clinical Characteristics

#### 3.2.1. Tinnitus Prevalence Cohort

Among the 90,681 patients with tinnitus with sufficient history and follow-up, 53.7% were female, 66.8% were White, and the mean age was 59.8 (SD: 14.3) years (Table 1). The mean CCI score was 2.6 (SD: 2.2) and the most common CCI disorders were chronic pulmonary disease (18.3%) and diabetes (16.6%). Over half (52.5%) of patients with tinnitus were aged 50–70 years.

#### 3.2.2. Tinnitus Incidence Cohorts

For the No-TNFα and Yes-TNFα cohorts, 64.8% and 66.3%, respectively, were female, 61.7% and 65.8% were White, the mean ages were 56.2 (SD: 17.5) and 53.2 (14.7) years, and the mean CCI scores were 2.6 (2.4) and 2.1 (1.8) (Table 2). The highest proportion of patients were aged 60–69 years in the No-TNFα cohort and 50–59 years in the Yes-TNFα cohort. In both cohorts, the most common CCI disorder was rheumatic diseases (No-TNFα: 28.6%, Yes-TNFα: 49.2%), followed by chronic pulmonary disease (19.6% and 11.4%) and diabetes (16.3% and 11.3%). Prior to matching, the Yes-TNFα cohort was, on average, younger that the No-TNFα cohort (53.2 vs. 56.2 years), had a longer follow-up (1577.3 vs. 1418.8 days), and had lower prevalence of most CCI comorbidities with the exception of rheumatic disease (all SMD > 0.25).

The demographic and clinical characteristics of the other cohorts are detailed in Appendix A. After matching, all cohorts were comparable on sex, race, age distribution, and baseline comorbidities.

### 3.3. Tinnitus Incidence According to Anti-TNFα Therapy

Patients with autoimmune conditions without tinnitus were evaluated for rates of tinnitus development, comparing those who did and did not receive anti-TNF-α therapy. After matching, 136 (1.3%) of the No-TNFα cohort and 173 (1.7%) patients in the Yes-TNFα cohort were diagnosed with tinnitus during the study period; the mean time to tinnitus diagnosis from index was 1053.7 (SD: 782.0) and 1179.9 (952.6) days, respectively. There were no significant associations observed between anti-TNFα treatment and risk of tinnitus, overall or stratified by age group. Specifically, compared with patients not treated with anti-TNFα, the hdPS-adjusted HR for incident tinnitus among those treated with anti-TNFα was 1.15 (0.92, 1.44) (Figure 1). When conducting this comparison between patients aged 30–50 years and 51–70 years, the hdPS-adjusted HRs for incident tinnitus were 0.85 (95% CI: 0.58, 1.23) and 1.16 (0.89, 1.51), respectively (Figure 2).

The incidence of tinnitus was also compared between patients with autoimmune disorders treated with different types of anti-TNFα therapies—AB or FP—and with patients who did not receive anti-TNFα therapy. After matching, 2.1% (*n* = 50) of the TNFα-FP cohort and 1.5% (*n* = 146) of the TNFα-AB cohort were diagnosed with tinnitus during the study period; the mean time to tinnitus diagnosis from index was 1135.4 (SD: 838.5) and 1063.1 (900.5) days, respectively. There were no significant associations between the type of anti-TNFα therapy and the risk of tinnitus (hdPS-adjusted HR [95% CI]: 0.79 [0.51, 1.22]) (Figure 3). Additionally, there were no significant associations with the risk of tinnitus when comparing patients with no anti-TNFα therapy use with those who used either AB (hdPS-adjusted HR [95% CI]: 1.00 [0.79, 1.28]) or FP (1.13 [0.79, 1.62]) anti-TNFα therapy (Appendix A).

### 3.4. Sensitivity Analyses

Compared to patients with autoimmune disorders and no anti-TNFα use, there were no significant associations between anti-TNFα therapy and tinnitus when restricting the Yes-TNFα population to those with ≥6 (hdPS-adjusted HR [95% CI]: 0.96 [0.69, 1.32]) or ≥12 months (1.03 [0.71, 1.50]) of anti-TNFα (Appendix A), or to only patients with RA (1.16 [0.88, 1.53]) (Appendix A).

## 4. Discussion

To our knowledge, this is the first study to examine the association between the use of anti-TNFα therapies and tinnitus incidence among adults with autoimmune disorders. Using a large US healthcare database, we matched patients with autoimmune conditions to reduce confounding and utilized rapid machine learning models such as hdPS to perform large-scale cohort studies while controlling for baseline confounders. The results indicated that there were no significant associations between anti-TNFα inhibitor use and tinnitus incidence in this patient population overall, or among higher risk age groups or patients using different types of anti-TNFα therapies. The results of the sensitivity analyses among patients with longer duration of anti-TNFα therapy or with the most prevalent autoimmune condition (RA) were consistent with the main findings.

Approximately 5% of patients with tinnitus in the EHR database had autoimmune disorders, a higher prevalence than typically reported in the general US population (3%) [35,36]. Tinnitus is not a known adverse event associated with anti-TNFα therapy and is not reported as occurring in their pivotal clinical trials or in their FDA prescribing information. However, patients with autoimmune disorders, particularly RA [37,38], are at higher risk of audio-vestibular symptoms due to autoimmune inner ear disease (AIED) [39,40]. In AIED, inflammation results in immune cells attacking the inner ear, leading to auditory deafferentation and peripheral auditory dysfunction manifesting as hearing loss, dizziness, and/or tinnitus [40]. AIED is typically treated with corticosteroids, although anti-TNFα has also been investigated. A small (*n* = 20) randomized trial found systemic etanercept to be no more effective than placebo [41], although transtympanic application of infliximab or golimumab resulted in hearing improvement in some patients [42,43]. Two case studies reported improvement of AIED-related hearing loss with adalimumab in a patient with CD [44] and another with RA [37]. Conversely, a case study implicated adalimumab in the hearing loss of a patient with arthritis and another with inflammatory spondylarthritis, and the latter’s symptoms resolved following cessation of adalimumab [45].

Tinnitus is a subjective disorder without objective clinical signs, presenting challenges for accurate estimation of its prevalence. Estimates of US tinnitus prevalence have varied widely (5–25.3% of adults), although there is a consistent finding of lower prevalence of severe tinnitus (~2%) [1,46,47,48]. To our knowledge, this is the first US healthcare claims or EHR analysis to assess the prevalence of clinically recorded tinnitus among adults, estimated at 0.55% during 2010–2021. This lower estimate compared to prior US-based epidemiological studies can be attributed to differences in the method of assessment—use of diagnosis codes in healthcare claims, thus requiring a healthcare encounter—compared to prior studies using patient-report surveys. For example, Bhatt et al. used data from the 2007 National Health Interview Survey to estimate a tinnitus prevalence of 9.6%, reflecting adults reporting any experience of tinnitus in the 12 months preceding the survey, but only 7.2% considered it a “big/very big” problem and less than half (49.4%) reported discussing tinnitus with a physician [4]. Further, while tinnitus can affect people of any age, prevalence and severity is correlated with increasing age [4,47,48,49]. “Bothersome” tinnitus increases up to 65–74 years, after which it becomes independent of age or decreases slightly [46]. In this study, the mean age of patients with tinnitus was ~60 years, older than prior reports (e.g., 53 years in Bhatt et al. [4]). This suggests that the current cohort represented those with severe/bothersome tinnitus who are both older and sought out diagnosis/treatment, explaining the comparatively lower prevalence. Additionally, non-US studies using EHR or claims data also reported lower prevalence, such as a South Korean study reporting that the 10-year national tinnitus prevalence was 1.44%, described as “clinically significant tinnitus” [50].

There are several explanations for the lack of an association between use of anti-TNFα inhibitors and tinnitus in this study population despite promising results in animal models. Tinnitus is a condition with complex and multifactorial etiology, pathophysiology, and clinical characteristics. Thus, not all emergent tinnitus may be related to neuroinflammation or TNFα-mediated pathogenesis, particularly in an older cohort at risk of presbycusis. Further, systemic anti-TNFα therapy at dosages and modalities approved for autoimmune disorders may not be efficacious to prevent cochlear damage. Application directly to cochlear fluids via the round window may produce different therapeutic effects. Future well-controlled studies may determine whether the timing of tinnitus onset or the method of anti-TNFα delivery are key factors in the treatment effect.

The results of this study are subject to several limitations, some of which are common to retrospective studies using EHR data. First, patients were identified and categorized based on diagnosis and treatment codes. Although there is a possibility of misclassification, the nationwide scope of the database helps reduce this error. The burden of tinnitus may not be fully captured in healthcare visits or diagnostic coding. Thus, the current prevalence/incidence estimates of tinnitus likely reflect patients with severe symptoms who seek out/receive diagnosis. Nevertheless, the results are meaningful as the first US-based EHR or claims database study to report the prevalence of tinnitus for which patients received a diagnosis. Second, information on the severity of tinnitus was not available in the EHR data. The availability of patient information was dependent on enrollment with insurers covered by the database. Similarly, patients’ blood serum or tissue levels of TNFα, or audiological data, were not available in the database. A sensitivity analysis was conducted to include patients that stayed on anti-TNAα for at least 6 or 12 months, which would be inferred to include patients with therapeutic levels of their anti-TNAα therapy. In acknowledgment of the important link between hearing loss and tinnitus, we analyzed two additional cohorts structured by age (30–50 and 51–70 years), selected due to the higher likelihood of presbycusis, and potentially related tinnitus among older age groups. The results of these analyses were consistent with those of the main analysis. Finally, the hdPS model did not exclude other sources of potential bias such as measurement error and residual error from unmeasured confounders.

## 5. Conclusions

This study showed that, after adjusting for potential confounders, anti-TNFα therapy was not associated with the incidence of tinnitus in patients with autoimmune conditions. Given the myriad comorbid factors that contribute to tinnitus and the heterogenous etiopathogenesis of the disorder, multiple pharmaceutical targets may be needed to reduce the disease burden.

## Figures and Tables

**Figure 1 jcm-12-01935-f001:**
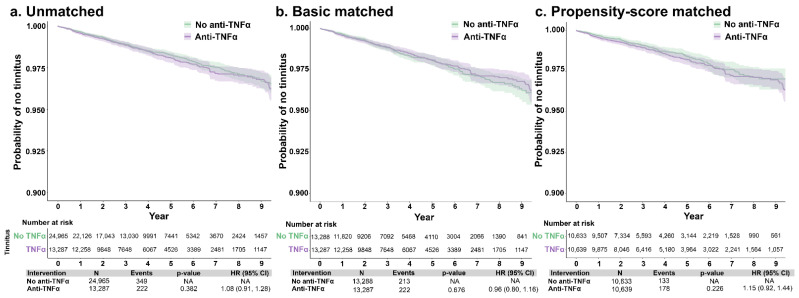
Unmatched (**a**), basic-matched (**b**), and propensity score-matched (**c**) comparisons of tinnitus incidence between patients with autoimmune disorders, by use of anti-TNFα therapy or no anti-TNFα therapy. The N of patients with anti-TNFα selected for the unmatched analysis was 25,000. When analyzing the incidence of tinnitus, the program encountered 35 missing values that were excluded. Abbreviations: anti-TNFα, tumor necrosis factor-alpha inhibitor; CI, confidence interval; HR, hazard ratio; NA, not applicable.

**Figure 2 jcm-12-01935-f002:**
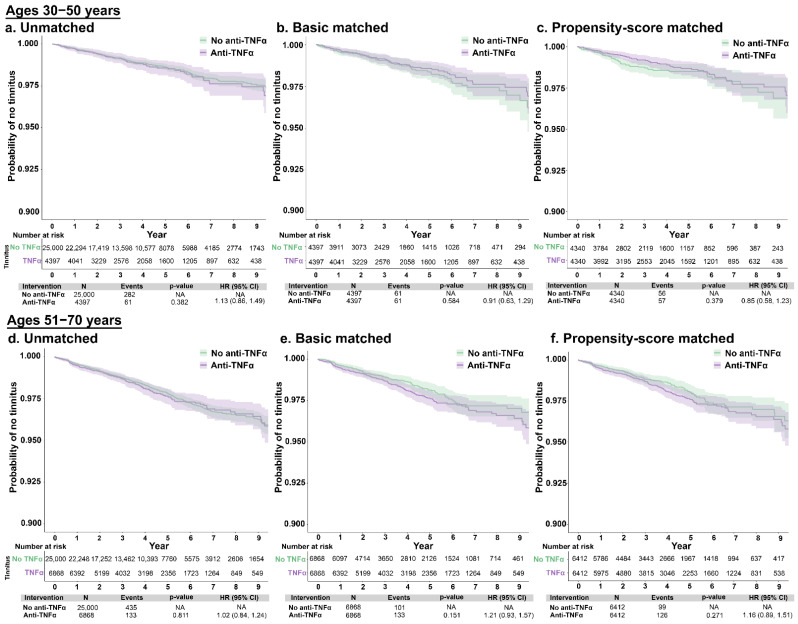
Unmatched, basic-matched, and propensity score-matched comparisons of tinnitus incidence between patients with autoimmune disorders aged 30–50 years (**a**–**c**) and 51–70 years (**d**–**f**), by use of anti-TNFα therapy or no anti-TNFα therapy. Abbreviations: anti-TNFα, tumor necrosis factor-alpha inhibitor; CI, confidence interval; HR, hazard ratio; NA, not applicable.

**Figure 3 jcm-12-01935-f003:**
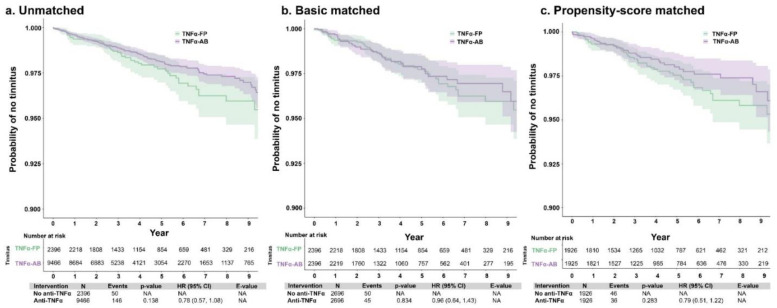
Unmatched (**a**), basic-matched (**b**), and propensity score-matched (**c**) comparisons of tinnitus incidence between patients with autoimmune disorders using anti-TNFα therapy, by therapy type (monoclonal antibody [TNF-AB] or fusion protein [TNF-FP]). Abbreviations: AB, monoclonal antibody; anti-TNFα, tumor necrosis factor-alpha inhibitor; CI, confidence interval; FP, fusion protein; HR, hazard ratio; NA, not applicable.

**Table 1 jcm-12-01935-t001:** Demographic and clinical characteristics of adults (age 18–89 years) with tinnitus in the EHR database during 2010–2021.

	Patients in the EHR with Tinnitus N = 155,091
N with minimum history ^a^ Mean pre-index days [SD]	90,681 1953.6 [1797.2]
Duration of follow-up, mean [SD] days	1280.2 [903.0]
Sex, n (%) female	48,661 (53.7%)
Age, years	
Mean [SD]	59.8 [14.3]
Distribution, n (%)	
18–29	3613 (4.0%)
30–39	5664 (6.2%)
40–49	11,000 (12.1%)
50–59	21,495 (23.7%)
60–69	26,114 (28.8%)
70–79	17,102 (18.9%)
80–89	5693 (6.3%)
Race/ethnicity, n (%)	
White	60,543 (66.8%)
Black	5969 (6.6%)
Asian	1002 (1.1%)
Other ^b^	23,167 (25.5%)
CCI	
Mean [SD] score	2.6 [2.2]
Component disorders, n (%)	
Malignancy	5211 (5.7%)
Metastatic solid tumor	272 (0.3%)
Diabetes	15,030 (16.6%)
Diabetes with complications	4495 (5.0%)
Congestive heart failure	2991 (3.3%)
Myocardial infarction	1011 (1.1%)
Peripheral vascular disease	4983 (5.5%)
Chronic pulmonary disease	16,613 (18.3%)
Cerebrovascular disease	5737 (6.3%)
Dementia	622 (0.7%)
Hemiparaplegia	379 (0.4%)
Mild liver disease	3435 (3.8%)
Severe liver disease	127 (0.1%)
Renal disease	5382 (6.0%)
Peptic ulcer disease	948 (1.1%)
Rheumatic disease	2633 (2.9%)
HIV	171 (0.2%)

Abbreviations: CCI, Charlson Comorbidity Index; EHR, electronic health records; HIV, human immunodeficiency virus; SD, standard deviation. Notes: ^a^ Demographic and clinical characteristics are reported among patients who met the inclusion criteria for history (90 days) and follow-up (180 days) in the EHR database. ^b^ ‘Other’ includes all other race/ethnicities as well as when this information was missing.

**Table 2 jcm-12-01935-t002:** Demographic and clinical characteristics of patients with autoimmune disorders who did or did not receive anti-TNFα therapy (Yes-TNFα and No-TNFα cohorts), before and after propensity score matching (2010–2021).

	Autoimmune Disorders Cohorts ^a^			
	Before Matching		After Matching	
	No-TNFα N = 25,000 ^b^	Yes-TNFα N = 13,293	No-TNFα N = 10,645	Yes-TNFα N = 10,645
Duration of follow-up, mean [SD] days	1418.8 [985.1]	1577.3 [1044.2]	1416.3 [968.3]	1653.5 [1072.3]
Sex, n (%) female	16,211 (64.8%)	8816 (66.3%)	6984 (65.6%)	6952 (65.3%)
Age, years				
Mean [SD]	56.2 [17.5]	53.2 [14.7]	53.4 [17.4]	53.4 [14.7]
Distribution, n (%)				
18–29	2566 (10.3%)	1046 (7.9%)	1332 (12.5%)	843 (7.9%)
30–39	2524 (10.1%)	1567 (11.8%)	1316 (12.4%)	1234 (11.6%)
40–49	3269 (13.1%)	2528 (19.0%)	1591 (14.9%)	1950 (18.3%)
50–59	4998 (20.0%)	3500 (26.3%)	2197 (20.6%)	2821 (26.5%)
60–69	5471 (21.9%)	2989 (22.5%)	2189 (20.6%)	2439 (22.9%)
70–79	4339 (17.4%)	1392 (10.5%)	1457 (13.7%)	1136 (10.7%)
80–89	1833 (7.3%)	271 (2.0%)	563 (5.3%)	222 (2.1%)
Race/ethnicity, n (%)				
White	15,421 (61.7%)	8752 (66.8%)	6855 (64.4%)	6799 (63.9%)
Black	1953 (7.8%)	826 (6.2%)	697 (6.5%)	692 (6.5%)
Asian	204 (0.8%)	105 (0.8%)	87 (0.8%)	88 (0.8%)
Other ^c^	7422 (29.7%)	3610 (27.2%)	3006 (28.2%)	3066 (28.8%)
CCI				
Mean score [SD]	2.6 [2.4]	2.1 [1.8]	2.2 [2.2]	2.1 [1.8]
Component disorders, n (%)				
Malignancy	1129 (4.5%)	249 (1.9%)	367 (3.5%)	216 (2.0%)
Metastatic solid tumor	58 (0.2%)	11 (0.1%)	12 (0.1%)	10 (0.1%)
Diabetes	4086 (16.3%)	1507 (11.3%)	1478 (13.9%)	1201 (11.3%)
Diabetes w/complications	1093 (4.4%)	321 (2.4%)	350 (3.3%)	255 (2.4%)
Congestive heart failure	987 (4.0%)	183 (1.4%)	316 (3.0%)	151 (1.4%)
Myocardial infarction	271 (1.1%)	78 (0.6%)	82 (0.8%)	66 (0.6%)
Peripheral vascular disease	1465 (5.9%)	268 (2.0%)	457 (4.3%)	236 (2.2%)
Chronic pulmonary disease	4893 (19.7%)	1520 (11.4%)	1851 (17.4%)	1214 (11.4%)
Cerebrovascular disease	1294 (5.2%)	273 (2.1%)	388 (3.6%)	227 (2.1%)
Dementia	192 (0.8%)	16 (0.1%)	49 (0.5%)	11 (0.1%)
Hemiparaplegia	99 (0.4%)	18 (0.1%)	37 (0.4%)	11 (0.1%)
Mild liver disease	833 (3.3%)	290 (2.2%)	313 (2.9%)	234 (2.2%)
Severe liver disease	55 (0.2%)	10 (0.1%)	24 (0.2%)	9 (0.1%)
Renal disease	1406 (5.6%)	305 (2.3%)	435 (4.1%)	249 (2.2%)
Peptic ulcer disease	344 (1.4%)	121 (1.0%)	118 (1.1%)	112 (1.1%)
Rheumatic disease	7152 (28.6%)	6538 (49.2%)	3155 (29.6%)	5012 (47.1%)
HIV	49 (0.2%)	7 (0.1%)	17 (0.2%)	5 (0.1%)

Grey highlight indicates characteristics with standardized mean difference >0.25 between groups. Patients in the two cohorts were propensity score-matched on sex, age, race/ethnicity, and CCI score. Abbreviations: CCI, Charlson Comorbidity Index; EHR, electronic health records; HIV, human immunodeficiency virus; SD, standard deviation; anti-TNFα, tumor necrosis factor-alpha inhibitor. Notes: ^a^ Demographic and clinical characteristics are among patients who met the inclusion criteria for history (90 days) and follow-up (180 days) in the EHR database. ^b^ The cohort was comprised of 25,000 randomly selected patients with no anti-TNFα use. ^c^ ‘Other’ includes all other race/ethnicities as well as when this information was missing.

## Data Availability

The authors attest that all study data are included in the article or online Appendix A.

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
