# Peer review of "Associations of Tinnitus Incidence with Use of Tumor Necrosis Factor-Alpha Inhibitors among Patients with Autoimmune Conditions"

_jcm, 2023, doi:10.3390/jcm12051935_

Round 1

Reviewer 1 Report

Authors evaluated whether the incidence of tinnitus among individuals with autoimmune conditions for which anti-TNFα are indicated differs between those who were and were not treated with anti-TNFα therapy, using a US electronic health records (EHR) database. Authors concluded that, after adjusting for potential confounders, anti-TNFα therapy was not associated with the incidence of tinnitus in patients with autoimmune conditions. This study is very interesting and valuable, however there are a few concerns to publish in current version.

Major concerns

1. The levels of TNFαin blood serum or tissue were not shown. Is it really related between TNFα levels and incidence? If this hypothesis is not correct, this study is meaningless.

2. Authors referred only the incidence of tinnitus, but the cure of tinnitus is important clinically.

3. This study lacked the information of hearing levels.

4. Please show the side effect of anti-TNFα. Is there no side effect (tinnitus) in anti-TNFα?

5. Authors should show whether autoimmune diseases was improved by anti-TNFα therapy. It might be related with tinnitus.

Minor concerns

1. The word size in Figures were too small. Please enlarged.

Reviewer 2 Report

This study is evaluated whether the incidence of tinnitus among individuals with autoimmune conditions for which anti-TNFα are indicated differs between those who were and were not treated with anti-TNFα therapy. As a retrospective cohort study has been evaluated.

Hearing loss is a crucial factor for the incidence of tinnitus. Did the authors make an evaluation in terms of hearing loss? Because there is no audiological evaluation in the study. It has been evaluated in many comorbidities that cause hearing losses. Additional cohorts have a presbycusis assessment. However, if there is a tinnitus due to hearing loss, it is always can be difficult to evaluate the effectiveness of anti-TNFα therapy.

Round 2

Reviewer 1 Report

All concerns were corrected rightly.